# Levels of Furaneol in Msalais Wines: A Comprehensive Overview of Multiple Stages and Pathways of Its Formation during Msalais Winemaking

**DOI:** 10.3390/molecules24173104

**Published:** 2019-08-27

**Authors:** Li-Xia Zhu, Meng-Meng Zhang, Zheng Liu, Yin Shi, Chang-Qing Duan

**Affiliations:** 1Production and Construction Group, Key Laboratory of High-Quality Agricultural Product Extensive Processing in Southern Xinjiang, College of Life Science, Tarim University, Alar, Xinjiang 843300, China; 2Key Laboratory of Viticulture and Enology, Ministry of Agriculture and Rural Affairs, Beijing 100083, China; 3Center for Viticulture and Enology, College of Food Science and Nutritional Engineering, China Agricultural University, Beijing 100083, China

**Keywords:** furaneol, furaneol formation, Msalais, Maillard reaction, biosynthesis, furaneol glucoside

## Abstract

4-Hydroxy-2,5-dimethyl-3(2*H*)-furanone (furaneol) is present in food. It has a caramel-like flavor, which affects the quality of food, and is formed via multiple pathways. Msalais is a traditional wine fermented from boiled local grape juice in Xinjiang (China). It has a strong caramel odor, which suggests high furaneol content. Furaneol formation during Msalais-making had not been investigated to date. Here, high-performance liquid chromatography and different fermentation models of Msalais-making were used to investigate the furaneol content and formation during Msalais-making. The furaneol content of Msalais is high, between 27.59 ± 0.493 mg/L and 117.6 ± 0.235 mg/L. It is formed throughout the entire Msalais-making process. The formation pathways include the Maillard reaction and chemical hydrolysis of bound furaneol during grape juice concentration; enzymatic release and/or chemical acidic hydrolysis of furaneol glucosides, and biosynthesis from Maillard products and d-fructose-1,6-diphosphate during fermentation; chemical transformation of Maillard products at room temperature (16–25 °C) and hydrolysis of furaneol glucosides during storage. Importantly, furaneol is formed by an efficient biotransformation of Maillard products. These findings suggest that furaneol content can be used as an important indicator of wine quality, and could be controlled by controlling the grape quality, grape juice concentration, fermentation, and wine storage.

## 1. Introduction

4-Hydroxy-2,5-dimethyl-3(2*H*)-furanone (furaneol) is a compound with a strong caramel-like flavor, present in food. It was first reported as a product of the Maillard reaction in 1960 [1]. It has been since identified in a variety of fruits (e.g., pineapple, strawberry, raspberry, and grape) [2,3,4,5] and vegetables (e.g., tomato, potato, leek, and onion) [6,7,8], and foods prepared from plants by heating or without heating (e.g., nonalcoholic beverages, wine, cooked potato, roasted almond, popcorn, roasted coffee, and green tea) [2,9,10,11,12,13]. It was also detected in cooked meat and milk products, including roast beef, human breast milk, and powdered and cooked milk [14,15,16,17]. Further, furaneol appears to form easily in heated and fermented foods, such as soy sauce, beer, bread, Baijiu, and cheddar cheese [12,18,19,20,21]. 

Dilute solutions of furaneol have a strawberry flavor, while furaneol concentrates have a caramel-like flavor [22]. Due to its pleasant taste and low odor thresholds (0.03 mg for taste and 0.1 mg/L water for odor) [12], furaneol is widely used as a flavoring agent in jams, jellies, ice cream, alcoholic drinks, and sweets [23,24]. Further, the compound has some physiological activity, e.g., it protects the human erythrocyte membranes and low-density lipoprotein against iron-induced oxidative modifications, effectively inhibits hyperpigmentation, and shows anti-infective activity during microbial infections in human [24].

The pronounced caramel-like flavor of furaneol and other 4-hydroxy-3(2*H*)-furanones is associated with the planar enol-oxo group of a cyclic dicarbonyl derivative that forms strong hydrogen bonds with the adjacent 4-hydroxy group [24]. On an industrial scale, furaneol is produced from l-rhamnose by heating [12]. Furaneol may also be formed via chemical transformation in model solutions containing d-fructose-1,6-diphosphate (FDP) and nicotinamide adenine dinucleotide phosphate [NAD(P)H] [25]. Further, furaneol naturally forms at the heating stage of food processing, via the Maillard reaction [26]. At room temperature, it is biosynthesized in fruit, and by yeast and bacteria, most probably via different pathways [24]. For instance, furaneol and its glucoside derivatives in strawberry fruit are synthesized from d-fructose and d-fructose 6-phosphate [27,28] by *Fragaria* × *ananassa* quinone oxidoreductase (FaQR) and *F.* × *ananassa* enone oxidoreductase (FaEO) (furaneol formation), and glucosyl transferase and malonyl transferase (glucoside derivative formation) [24,29]. On the other hand, the yeast *Zygosaccharomyces rouxii* transforms exogenous FDP to furaneol (27). *Z. rouxii* and bacteria (*Lactococcus lactis subsp. cremoris*) also produce furaneol in vitro, in a medium prepared by damp-heat sterilization [24,25,30]. The bio-formation pathways of furaneol by microbes are complicated and have not been fully delineated [24]. As furaneol is widely present in agricultural products and foods, and is formed via complicated pathways in different food matrices, it is difficult to control the quality of food containing high levels of furaneol. 

Msalais is a traditional wine produced by spontaneous fermentation of boiled grape juice by the Uygur people in the Xinjiang Uygur Autonomous Region (XUAR) in China. The Msalais tradition and Msalais-producing technology have been passed down from generation to generation in the A’Wati region in XUAR, the only main production region of Msalais. Since 2007, Msalais has been protected as an intangible human cultural heritage. The flavor characters of Msalais and strong caramel odor are different from those of other wines primarily because it is produced from a local grape, *Vitis vinifera* Hetianhong, often planted in the courtyards in Southern Xinjiang, with a 1600-year growth history and a neutral aroma [31]. Another explanation of the special characteristics of Msalais is the unique winemaking technology: local grape juice is boiled (to concentrate the sugar, extract the color, and produce flavor via the Maillard reaction), and the cooled grape juice is subsequently naturally fermented into Msalais within 45 days, and stored for 1–3 years at room temperature [32,33]. The caramel flavor is dominant in Msalais, however, its intensity varies from pleasant to very strong; in extreme cases, it is too strong, which diminishes the aromatic and flavor complexity of the wine [34]. Considering the strong caramel odor of furaneol, we hypothesized that the wine quality may potentially be associated with a variable furaneol content. Hence, in the current study, the furaneol content and formation during Msalais-making were explored, to create a theoretical basis for improving the quality of wine by controlling its furaneol content. 

## 2. Results

### 2.1. Furaneol Content of Msalais

As shown in Table 1, the °Brix, alcohol, and pH values of the 13 Msalais wines supplied by different producers were significantly different (*p* < 0.05). The furaneol content was between 27.59 ± 0.49 mg/L and 117.60 ± 0.24 mg/L, and also significantly different (*p* < 0.05) between samples. To evaluate the furaneol odor intensity in Msalais, the odor activity values (OAVs) of furaneol in Msalais were calculated by dividing the contents of furaneol in wines by the threshold value in water (0.1 mg/L) [20] and in 10% hydroalcoholic solution at pH 3.2 (5 μg/L) [35]. For all tested wines, the OAV of furaneol exceeded 1, reaching hundreds to thousands in some cases (Table 1). These observations indicated that furaneol plays an important role not only in the aroma but also in the flavor of Msalais.

### 2.2. Formation of Furaneol during Msalais-Making

An increase of furaneol content in the different winemaking experiments with time was observed, as shown in Figure 1. 

Considering sample type, grape juice (Sgj, Ogj, and mgj) contained less than 2 mg/L furaneol, which was less than that in other samples (Figure 1, Appendix A); heated simulated grape juice (sb) contained approximately 1.40 ± 0.14 mg/L furaneol (Appendix A); and the furaneol contents of Sgi, Ogj, mgj, and sb were significantly different (*p* < 0.05) (Appendix A). 

Further, the furaneol content increased during grape juice concentration (Figure 1a–d, Appendix A), up to 9.0–15 mg/L end concentration in bSgj120min, bOgj120min, bmgj90min, and bSgej120min. In sb, it was significantly lower (*p* < 0.05) than that in the boiled grape juice (bSgj, bOgj, bmgj, and bSgej) (Appendix A). 

During fermentation (day 0 to approximately day 45), the furaneol content increased more than 6.3–9.3 times compared with the content after grape juice concentration (bSgj120min, bOgj120min, bSgej120min, and bmgj90min), to over 84 mg/L in the approximately 45 days fermentations (fbSgj48d, dfbOgj48d, afbOgj48d, fbSgej48d, and fbmgj44d) (Figure 1a–d, Appendix A). In the sb model, the furaneol content increased up to 21.42 ± 0.51 mg/L in afsb48d and 25.54 ± 0.84 mg/L in dfsb48d (Figure 1e, Appendix A). That was 15.3 and 18.24 times higher, respectively, than furaneol content in sb (1.401 ± 0.129 mg/L). For the natural fermentation of grape juice, the furaneol content increased 6.02 times in nfSgj48d and 5.53 times in nfOgj48d compared with the furaneol content in Sgj and Ogj grape juice, accordingly (Figure 1f, Appendix A). The final Msalais wine, after 90–256 days of storage, contained over 100 mg/L furaneol and was significantly different between fbSgj256d, fbSgej256d, afbOgj245d, dfbOgj245d, and fbmgj90d (*p* < 0.05) (Figure 1a–d, Appendix A). Further, the furaneol content increased in the nf group, up to 23.32 ± 0.165 mg/L in nfgOj245d and 27.47 ± 0.07 mg/L in nfSgj256d (Figure 1f, Appendix A). During storage in the sb model, the furaneol content increased up to 43.83 ± 0.17 mg/L in afsb114d and 41.84 ± 0.37 in dfsb114d, which was significantly higher than that in the nf group (*p* < 0.05) (Figure 1e,f, Appendix A). Furthermore, furaneol content in samples fermented and stored for the same or similar periods of time, e.g., 90 days, was also significantly different (*p* < 0.05) and ranked, in decreasing order, as follows: fbSgej > fbOj > fbSgj > fbmj > fsb > nf (Appendix A). 

For fermentations using different fermenting starter cultures, the furaneol content in samples inoculated with a starter from the Daolang modern plant (df) was higher than that in samples inoculated with a starter from the Ahuizhang craft workshop (af), e.g., dfsb114d vs. afsb114d (Figure 1e, Appendix A), or dfbOjg245d vs. afbOjg245d (Figure 1c, Appendix A). 

These observations indicated the furaneol was produced at different stages of Msalais-making, mainly during fermentation and storage. More furaneol was produced by microbiological transformation of sb than of grape juice. The different starters used for fermentation also influenced the furaneol content of the final product.

### 2.3. Analysis of Furaneol Released by Hydrolysis from Furaneol Glucosides 

Furaneol was produced in the naturally fermented grape juice (Figure 1e). More furaneol was produced during the fermentation of boiled grape juice (bSgj120min, bSgej 120min, bmgj90min, and bOgj90min) (Figure 1a–d) than during the fermentation of sb (Figure 1e) and natural grape juice (Figure 1f). This indicated the presence of furaneol glucosides and derivatives in the grape juice. To verify this, furaneol was released by enzymatic or thermal acidic hydrolysis from the Sgj, Sgej, and bSgej120min extracts, and from the fbSgej13d and fbSgej48d fermentations, and after 90 days of storage (Figure 2). Slightly more furaneol was released from the Sgj extract than from the Sgej extract, but the content did not exceed 0.10 ± 0.08 mg/L and 0.18 ± 0.07 mg/L, respectively. On the other hand, the quantity of furaneol released from the bSgej120min extract (3.51 ± 0.13 mg/L released by enzymatic hydrolysis, and 2.97 ± 0.01 mg/L released by thermal acidic hydrolysis) was approximately 10 times and significantly higher (*p* < 0.05) than that released from Sgej and its subsequent fermentations. The content of furaneol released from the fermented extracts significantly decreased with the fermentation time (*p* < 0.05). Although different quantities of furaneol were released by enzymatic and thermal acidic hydrolysis, the content of furaneol released by either method showed the same trends in the analyzed samples.

## 3. Discussion

In the current study, the furaneol content and formation pathways were investigated during Msalais-making. The data indicated that Msalais contains unusually high levels of furaneol, and that the compound is formed as a result of a complex interplay of abiotic and biotic factors.

Msalais contains at least 27.59 ± 0.49 mg/L furaneol, which is considerably higher than the values reported for other alcoholic beverages to date [2]. The furaneol content in alcoholic beverages varies from trace amounts to 3.5 mg/L in wine (depending on the grape variety), 2.0–8.0 mg/L in beer, and 6.6 mg/L in strawberry wine [2]. A compound with OAV greater than 1 contributes to the wine aroma, and the higher the OAV, the greater the contribution [36]. Hence, the high OAV of furaneol in Msalais (exceeding 1) indicated its important contribution to flavor and, especially, aroma characters of Msalais. The highly variable furaneol levels in the sampled Msalais wines indicate varying quality of the wine.

The high content of furaneol in Msalais is an outcome of the winemaking process as a whole, with fermentation as the main stage of furaneol formation. The possible furaneol formation pathways during Msalais-making are presented in Figure 3. These pathways are based on the observations made in the current study and on the furaneol formation pathways reported earlier [24,26,27,37,38,39,40,41]. The data from the current study suggest that furaneol formation in Msalais involves the Maillard reaction and thermal hydrolysis of bound furaneol during grape juice concentration; enzymatic release and chemical acidic hydrolysis of bound furaneol during fermentation; biosynthesis from FDP and/or the Maillard products during fermentation and storage; and chemical transformation of FDP and/or the Maillard products at room temperature during storage.

Furaneol formation during grape juice concentration for Msalais-making could be ascribed to the Maillard reaction and chemical hydrolysis (Figure 3, red). In the absence or presence of amino acids, in a thermal system, hexoses, 6-deoxysugars, and pentoses can be degraded to furaneol via a major pathway that involves the 2,3-enolisation of sugar, β-elimination of water or an amino group, intra-molecular cyclisation, and dehydration [41]. In heat-processed foods containing hexoses and amino acids, furaneol always forms via the Maillard reaction, with acetylformoin, diacetylformoin, and dihydrodiacetylformoin as important intermediates [38,39,42]. As a major formation pathway, glucose is cleaved into acetaol and lactaldehyde; acetylformoin is then formed by idolization; and acetylformoin is reduced to furaneol [26,38]. Acetylformoin can also be obtained from FDP as a precursor, under pyrolytic conditions [12]. In the current study, furaneol formation in the heated stimulated grape juice (sb) and increased furaneol levels during grape juice concentration (bSgj, bOgj, bmgj, and bSgej samples) (Figure 1) provided direct evidence of furaneol formation via the Maillard reaction. 

Further, the occurrence of thermal hydrolysis of bound furaneol during grape juice concentration was indirectly confirmed. First, the furaneol content in the final concentrated grape juice (Figure 1a–d) was higher than that in the heated simulated grape juice (Figure 1f). Second, the increase of furaneol levels during the concentration of a mixture of grape juice and grape residue extracts (bSgej0min–120min) was higher than that observed during the concentration of samples containing only grape juice (bSgj0min–120min) (Figure 1a,b), even though both sample types were obtained from the same grape (Sj). Thermal acidic hydrolysis of bound glucosides to release furaneol was further evidenced by a successful identification of furaneol as one of the products of thermal acidic hydrolysis (100 °C for 1 h, pH 4.0) of the extracts of Sgj, Sgej, and bSgej120min samples (Figure 2). The increased levels of furaneol during grape concentration indicated that the bound furaneol mainly originated from grape residues (skin and/or seed) in these samples. Furaneol was also released by thermal hydrolysis during the concentration of the mixture of grape juice and grape residue extracts (Figure 2). 

Furaneol accumulates largely during the fermentation stage. The two pathways of furaneol formation, namely, the release of bound furaneol by chemical or enzymatic hydrolysis (Figure 3), are important contributors to the furaneol content. Furaneol and its derivatives are present in different grape species [35,43,44,45,46]. In the current study, the furaneol content increased during natural fermentation of grape juice without pre-heating, which could be attributed to the enzymatic release and/or chemical acidic hydrolysis of the bound furaneol (Figure 1e, Figure 2). Specific glucosides of furaneol in grape are present in the form of furaneol glucosides [4]. While furaneol glucosides present in strawberry, such as furaneol-glucuronide and furaneol-malonylated glucoside, have not yet been identified in grape [4,7,47], grape contains furaneol aglycon [4]. The glucosides have a direct linkage of furaneol to a β-d-glucose moiety [2,7,24,48]. In the current study, Msalais obtained from concentrated Hetianhong grape juice or grape juice containing grape residue extracts contained more furaneol than Msalais from concentrated Munage grape juice (Figure 1a–d). This indicated that Hetianhong grape juice might contain more bound furaneol than Munage grape juice. Indeed, Hetianhong grape juice contained more free furaneol (Figure 1a,c) than Munage grape juice (Figure 1d). This indirectly indicated that Hetianhong grape contains more bound furaneol than Munage grape. However, additional experiments are required to confirm this. 

During Msalais fermentation, furaneol was generated via biological and chemical pathways from FDP and (highly likely) the Maillard products (Figure 3). This was evidenced by furaneol formation during fermentation in the sb model, a heated artificial synthetic medium containing hexoses and amino acids (Figure 1f). Although the complete pathway of furaneol formation in yeast has not yet been delineated, the following important points should be considered: the carbons of furaneol originate exclusively from exogenous FDP [27]; FDP is biotransformed to furaneol by the yeast *Z. rouxii* and in strawberry fruit [27]; and FDP is chemically transformed to 1-deoxy-2,3-hexodiulose-6-phosphate via 2,3-enolization and then enzymatically to furaneol by *Z. rouxii* [25]. Further, FDP is also enzymatically transformed into dihydroxyacetone phosphate by exogenous biphosphate aldolase and triose phosphate isomerase; the product of the reaction can react with lactaldehyde, via an aldol condensation, to produce 6-deoxyhexose; and furaneol is subsequently formed during heating at 80 °C for 20 h with piperidine in an acidic-ethanol solution [40]. Furaneol glucoside is biosynthesized from 6-deoxyhexoses as precursors of the furan ring of furaneol in strawberry [49], but it is not known whether microbes could also biotransform 6-deoxyhexoses into furaneol. During Msalais-making, FDP would originate from the grape juice and/or would be released by yeast; aldolase would be secreted by yeast; and lactaldehyde would be produced by the Maillard reaction during grape concentration. Further, grape juice is acidic, and the wine contains at least 8% ethanol and has low pH (3.2–4.0). These observations suggest the existence of an enzymatic-chemical pathway of furaneol formation during yeast fermentation, and a potential alternative microbial transformation of 6-deoxyhexoses to furaneol. Currently, the knowledge of microbiological pathways of furaneol formation, especially biotransformation from the Maillard products, is limited. Theoretically, the yeast should be able to assimilate sugar fragments produced during the Maillard reaction more easily than hexoses. In the current study, the furaneol content increased during fermentation in the sb model in the absence of exogenous FPD, providing direct evidence for the biological formation of furaneol from the Maillard products, most likely involving acetol, acetylformoin, in addition to FPD. That is the first and unambiguous demonstration of the biotransformation of furaneol from Maillard products. In the future, fermentation experiments in the presence of the potential Maillard products as substrates, combined with isotope analysis, should be performed to verify the proposed pathway of furaneol formation.

Finally, during Msalais fermentation and storage at room temperature, furaneol might be formed via chemical transformation (Figure 3, green font). In addition to the biotransformation of FPD to furaneol, FDP can also be chemically converted to furaneol in the Maillard reaction, with or without heating [12], or in solutions containing FDP and NAD(P)H at room temperature (perhaps via mechanisms similar to those during heating), with acetylformoin as a key intermediate (Figure 3) [37]. Phosphate ions stimulate furaneol production via the Maillard reaction, and FDP is rapidly converted to furaneol under room-temperature conditions [12,50]. During Msalais-making, the concentration of phosphate ions is sufficient to allow the yeast to grow, and FDP and NAD(P) could be derived from the yeast lysate, with acetylformoin formed by the Maillard reaction. Therefore, furaneol could be formed by a chemical transformation during fermentation and during Msalais storage at room temperature. This could be confirmed in the future by experiments involving fermentation in the presence of these compounds (FDP, NAD(P), acetylformoin, etc.). 

Collectively, furaneol is formed via multiple pathways during Msalais-making (Figure 3). The contribution of these pathways to the furaneol content in Msalais is as follows, in increasing order: the isolated Maillard reaction (sb, 1.401 ± 0.13 mg/L furaneol), and chemical hydrolysis with the Maillard reaction upon heating (bSgj, bSgej, and bOgj samples, from 9.01 ± 0.06 mg/L to 14.625 ± 0.20 mg/L furaneol); microbial and chemical hydrolysis of furaneol glucosides (up to 27.468 ± 0.07 mg/L during natural fermentation of grape juice after 256 days) (Figure 1f); and microbial and chemical transformation of the Maillard products (sb, up to 43.83 ± 0.17 mg/L after 114 days (Figure 1e). All these pathways together contribute to the high and variable furaneol content in Msalais (Table 1), with the microbial and chemical transformation of the Maillard products as the main pathway. 

From a practical perspective, considering the high furaneol content of Msalais and the fact that furaneol is a strong caramel-like compound in alcoholic beverages [35], furaneol could be used as an important indicator for controlling the quality of wine. The higher the level of furaneol, the stronger the caramel odor of wine. However, while moderate furaneol content enhances the quality of wine, extremely high furaneol content and strong caramel odor diminishes the aromatic and flavor complexity of wine, impairing the wine quality. Standards for acceptable furaneol content in wine could be set. Although fermentation and storage are the main stages during which furaneol is generated, the grape quality and concentration intensity of the grape juice determine the species of compounds for furaneol formation during these stages (e.g., furaneol glucosides and Maillard intermediates). It is hence necessary to synergistically control the grape quality, concentration of grape juice, fermentation, and storage, e.g., by using grape with moderate furaneol glucoside content; concentrating the grape juice with or without extraction of grape residues; fermentation using selected starters and appropriate fermenting parameters; and storage with or without lees. A specific program to control the content of furaneol generated at different stages of winemaking should be carefully optimized for Msalais, focusing on the fermentation stage, and mainly considering the biotransformation of Maillard products and enzymatic hydrolysis of furaneol glucosides. Further, since all the currently known pathways of furaneol formation in the food matrix possibly occur during Msalais-making, Msalais production could be used as a model for researching the formation mechanisms of furaneol in food, e.g., for qualitative and quantitative identification of furaneol glucosides in specific grape varieties, and key intermediates during chemical and microbiological transformation from the Maillard products.

## 4. Materials and Methods 

### 4.1. Chemicals 

Methanol and acetonitrile (HPLC grade), and furaneol standards (≥98%) for the HPLC analysis were from Sigma-Aldrich (St Louis, MO, USA). For the simulated grape juice models, individual amino acids (biological grade), glucose (biological grade), fructose (biological grade), and salts (analytical grade) were purchased from Shanghai Yuan Ye Biotechnology Co., Ltd. (Shanghai, China). Amberlite XAD-2 (150 mg/6 mL) was purchased from Tianjin Boer Ajir Technology Co., Ltd. (Tijanjin, China). Double-distilled water was used in all experiments. Rapidase revelation aroma (AR 2000) from *Aspergillus niger* was purchased from Creative Enzymes (New York, NY, USA).

### 4.2. Msalais 

In the current study, different samples of Msalais wine (13 different Msalais wines) were obtained from the maximum number of producers, as feasible, from the Awati production region. Generally, the sampled wines were produced from ripe Hetianhong grape in 2017, and prepared by the two basic processes of grape concentration and spontaneous fermentation. The alcohol content in wine was at least 8%, with 7.0–14.3°Brix, and pH of 3.09–4.10 (Table 1). The Msalais samples were prepared in triplicate for analysis. 

### 4.3. Preparation of Models for Simulated Msalais-Making 

To analyze the multiple pathways of Msalais formations at the different stages of Msalais making (grape juice concentration, fermentation, and storage), the six models were designed as the follows: 

Model Sj1: ripe Hetianhong grapes (19°Brix) harvested on 17 September 2017 → cleaning → de-stalking → weighing (10 kg) → juice pressing → grape juice (Sgj) → concentration (samples bSgj0min–120min) → cooling down to 25 °C → natural fermentation (samples fbSgj1d–fbSgj41d) → Msalais storage (samples fbSgj45d–fbSgj256d).

Model Sj2: ripe Hetianhong grapes (19°Brix) harvested on 17 September 2017 → cleaning → de-stalking → weighing (10 kg)→ juice pressing → grape residues boiled with 2 L of water to 16°Brix → filtering into grape juice and fully mixing (Sgej) → concentration (samples bSgej0min–120min) → cooling down to 25 °C → natural fermentation (samples fbSgej1d–fbSgej41d) → Msalais storage (samples fbSgej48d–fbSgej256d).

Model Oj: Ripe Hetianhong grapes (22°Brix) harvested on 2 October 2017 → cleaning → de-stalking → weighing (20 kg) → dividing into four equal portions (10 kg each) → juice pressing → grape juice (Ogj) → concentration (samples bOgj0min–90min) → cooling down to 25 °C → inoculation with the df or af starters, as indicated → inoculated fermentation (samples afbOgj1d–afbOgj33d and dfbOgj1d–dfbOgj33d) → storage (samples afbOgj48d–afbOgj245d and dfbOgj48d–dfbOgj245d).

Model mj : ripe *V. vinifera* cv. Munage grape (20°Brix) harvested on 17 September 2017 → cleaning → de-stalking → weighing (10 kg) → juice pressing → grape juice (mgj) → concentration (samples bSgj0min–90min) → cooling down to 25 °C → natural fermentation (samples fbmgj1d–fbmgj33d) → Msalais storage (samples fbmgj44d–fbmgj90d).

Model nf: Sgj and Ogj grape juice (8 kg each) → natural fermentation (samples nfSgj1d–41d and nfOgj1d–41d) → storage (samples nfSgj48d–256d and nfOgj48d–245d).

Model sb: a synthetic grape juice solution [51] (220 g/L sugar with the glucose and fructose ratio 1:1, pH 3.5, 3 L) → dividing into two equal portions (1.5 L each) → boiled with stirring (800 rpm) to approximately 28°Brix in 30 min, using a multi-head temperature-controlled magnetic stirrer → cooling down to 25 °C (samples sb) → inoculation with the df or af starters, as indicated → inoculated fermentation (samples afsb1d–afsb33d and dfsb1d–dfsb33d) → storage (samples afsb48d–afsb114d and dfsb48d–dfsb114d).

In the six models, except for the sb model, the liquid concentrations were performed in a 20-L steel barrel on a gas stove, up to approximately 28°Brix in 90–120 min, with manual stirring. The fermentations from grape juice and their derivatives were performed in a 10-L Chinese ceramic jar with a water seal (cleaned and heated for 30 min by water at 80–90 °C in a 50-L steel barrel), sealed by boiled water. The fermentation of sb was performed in a 2-L flask (cleaned and heated for 30 min by water at 80–90 °C in a 50-L steel barrel), sealed by a plastic wrap. The natural fermentation liquids (200 mL) at the high-bubble stage from Daolang modern plant (df) or Ahuizhang craft workshop (af) were centrifuged (5000 rpm/min), and washed with sterile water three times to use as starters for the inoculated fermentations in the Oj model. Starters from 20 mL of af and df natural fermentation liquids for the sb model were prepared in the same manner. The fermentations in the other four models were all natural, i.e., without inoculation with starters. The duration of the fermentation in the six models was approximately 45 days, and proceeded at room temperature (20–25 °C). The liquids of the six models were stored in the same jar type (except for the sb model, which was performed in the same flask as that used for fermenting), up to 256 days at room temperature (16–20 °C). For the analysis, each model was prepared in duplicate and the samples were collected at the indicated time points, in triplicate (5 mL per replicate). All samples were stored in a refrigerator at –80 °C for furaneol tests. 

### 4.4. Enzymatic and Thermal Acidic Hydrolysis of Furaneol Glucosides

The quantity of furaneol released from glucosides during Msalais-making was analyzed in grape juice (Sgj) and in selected samples of the Sj2 simulated model (Figure 2), by enzymatic and thermal acidic hydrolysis. Before the hydrolysis, the samples were prepared as follows. A solid-phase extraction column Amberlite XAD-2 was activated by washing with 10 mL of methanol, followed by 10 mL of distilled water; 2 mL of sample was then loaded. Next, 2 mL of water was passed through the column to elute polar low-molecular weight compounds, such as sugars and acids; 5 mL of dichloromethane was then passed through the column to remove free aroma substances. The bound glucosides were eluted with 20 mL of methanol and collected in a 50-mL round-bottom flask. The eluent flow rate was maintained at 2 mL/min throughout the solid-phase extraction. The collected material was dried using a rotary vacuum evaporator at 30 °C for 30 min. For the enzymatic digestion, the dried material containing the bound glucosides was fully dissolved in 10 mL of citric acid buffer (2 M, pH 5), which was then transferred to two 10-mL centrifugation tubes (4.9 mL in each tube). Next, 100 μL of glycosidase AR 2000 (100 g/L) was added, and the enzymatic hydrolysis allowed to proceed for 16 h in a water bath at 40 °C. For the acid hydrolysis, the dried bound glucosides were fully dissolved in 10 mL of citric acid buffer (2 M, pH 4.0), and incubated for 1 h in a water bath at 100 °C. Each sample was prepared in triplicate. The resultant samples were analyzed by HPLC (Section 4.5).

### 4.5. HPLC Analysis

Furaneol content was analyzed using an LC-20AB Shimadzu Series HPLC (Shimadzu Technologies, Shanghai, China) equipped with a quaternary pump, an auto-sampler injection system, an Ultimate XB-C18 column (250 × 4.6 mm, 5 μm, Shimadzu, Japan), a degasser, a photo-diode array detector (PDA-M20A), and a UV/VIS detection set to 286 nm. The system was controlled using a Shimadzu Chem Station for Windows (Shimadzu Technologies). To detect furaneol, HPLC analysis was conducted, with methanol/0.5% formic acid solution (*v*/*v*) as the mobile phase, and the following methanol flow gradient: 15/85—2 min, 50/50—24 min, 100/0—27 min, 100/0—29 min, and 15/85—33 min. The operating conditions were as follows: flow rate of 0.8 mL/min; 33-min run; column temperature of 35 °C; and injection volume of 10 μL. A calibration curve (0.076–1200 mg/L, R^2^ = 0.9992) with 0.076 limit of quantification was prepared using a furaneol standard (Sigma-Aldrich). The samples were injected in duplicate. 

### 4.6. Statistical Analysis

The data were analyzed by one-way analysis of variance (ANOVA) with Tukey’s test (mean values comparison) to identify significant differences between samples by using SPSS 19.0 (IBM Corporation, Somers, NY, USA). The value of *p* < 0.05 was considered significant. Histograms of furaneol content in different models (mean ± SD from triplicate experiments) were also generated by using SPSS 19.0. Furaneol formation pathways were drawn by using ChemDraw Pro 17 (Cambridge Soft Corporation, Cambridge, MA, USA).

## 5. Conclusions

In the current study, the furaneol content of analyzed Msalais wines was shown to be high, between 27.59 ± 0.49 mg/L and 117.60 ± 0.24 mg/L. Furaneol is extensively formed during Msalais-making, mainly at the fermentation stage, and via multiple pathways. These pathways involve the Maillard reaction, chemical hydrolysis, enzymatic release of bound furaneol, and biosynthesis from the Maillard products. The presented findings might inform ways to improve the Msalais technology, e.g., by using furaneol as an important indicator of wine quality.

## Figures and Tables

**Figure 1 molecules-24-03104-f001:**
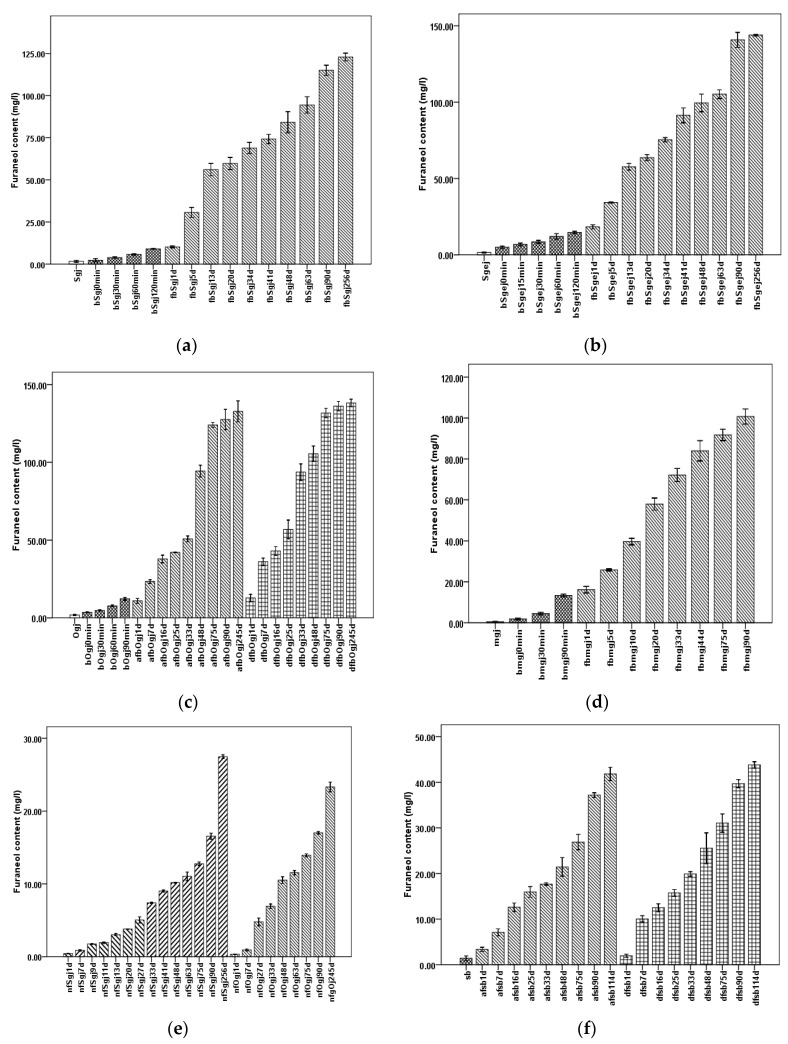
Changes of furaneol content during Msalais-making. The furaneol content was determined in six different models: (**a**) Sj1 model; (**b**) Sj2 model; (**c**) Oj model; (**d**) mj model; (**e**) sb model; and (**f**) nf model. Samples: sgj, grape juice from grape harvested in September 2017; Sgej, a mixture of sgj grape juice and liquid extracts (16°Brix) of grape residues obtained by adding water and boiling; Ogj, grape juice from grape harvested in October 2017; mgj, grape juice from Manaizi grape harvested in September 2017; sb, concentrate of a synthetic grape juice solution; bSgj0min–bSgj120min, bSgej0min–bSgej120min, bOgj0min–bOgj90min, and bmgj0min–bmgj90min indicate the different boiling times used to concentrate Sgj, Sgej, Ogj, and mj, respectively; fbSgj1d–fbSgj256d, fbSgej1d–fbSgej256d, and fbmgj1d–fbmgj90d indicate natural fermentation of cooled bsgj120min, bSgej120min, and bmgj90min on different fermenting days, respectively; afbOgj1d–afbOgj245d and dfbOgj1d–dfbOgj245d, and afsb1d–114d and dfsb1d-dfsb114d indicate the fermentation of cooled bOgj120min and sb, respectively, on different fermenting days, and inoculated using starters from Ahuizhang craft workshop (“a…”) and Daolang modern plant (“d…”); nfSgj1d–nfSgj256d and nfOgj1d–nfOgj245d indicate the natural fermentation of Sgj and Ogj, respectively, conducted for different times. The data are presented as the mean ± SD from three independent experiments, with three replicates each.

**Figure 2 molecules-24-03104-f002:**
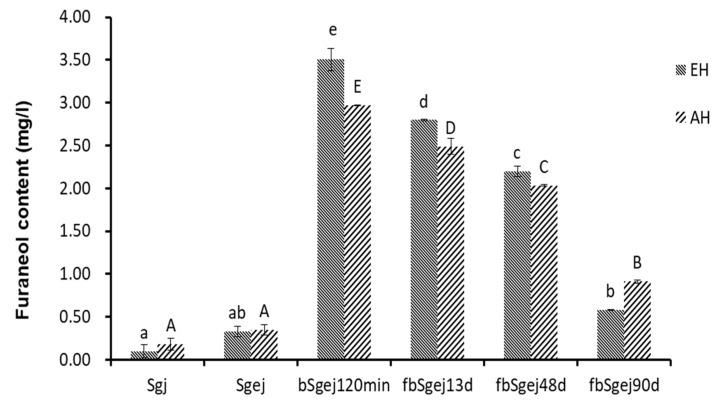
The quantity of furaneol released from furaneol glucosides by enzymatic hydrolysis (EH) (by glycosidase AR 2000, at 40 °C for 16 h, pH 5.0) and thermal acidic hydrolysis (AH) (100 °C for 1 h, pH 4.0). The quantity of released furaneol was determined by high-performance liquid chromatography (HPLC). The data are presented as the mean ± SD from three independent experiments, with three replicates each. Different lowercase or capital letters over the bars respectively indicate significant differences between the enzymatically hydrolyzed or thermal acidic hydrolyzed samples analyzed by one-way ANOVA and Tukey’s test at *p* < 0.05. Samples: sgj, grape juice from grape harvested in September 2017; Sgej, a mixture of sgj grape juice and liquid extracts (16°Brix) of grape residues obtained by adding water and boiling; bSgej120min indicates the concentrate of Sgej for 120 min; fbSgej13d, fbSgej48d, and fbSgej90d indicate natural fermentation of cooled bSgej120min on different fermenting days.

**Figure 3 molecules-24-03104-f003:**
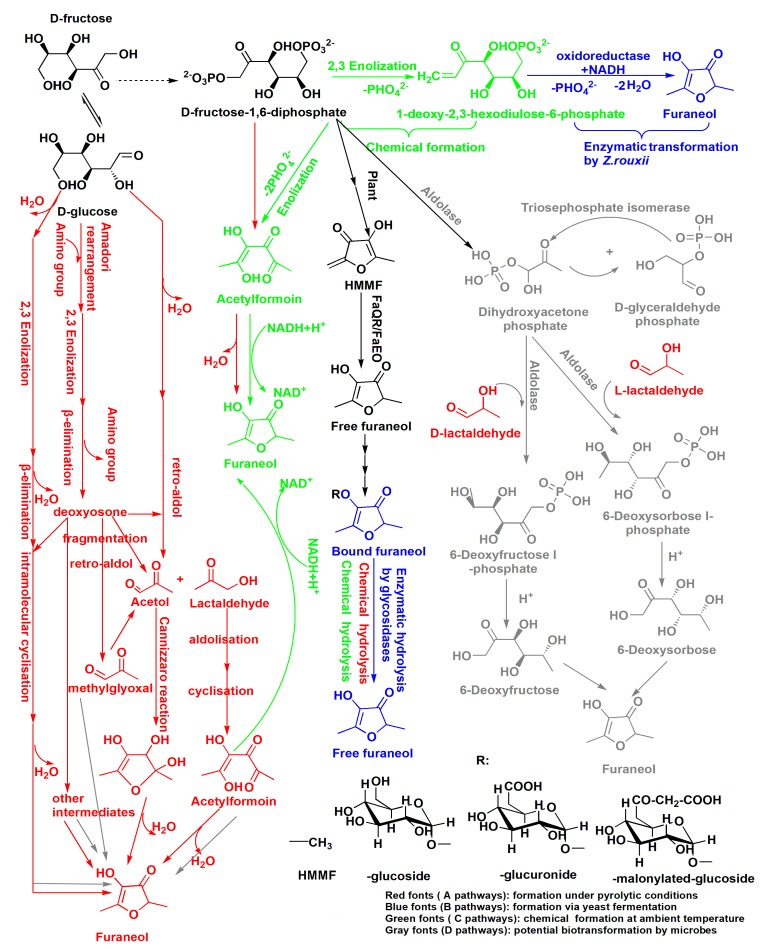
Multiple pathways of furaneol formation during Msalais wine making. FaQR: *Fragaria* × *ananassa* quinone oxidoreductase, FaEO: *F.* × *ananassa enone* oxidoreductase, HMMF: 4-hydroxy-5-methyl-3(2*H*)-furanone. The pathways were reported by Nashalian et al. [26], Dahlen et al. [27], Hauck et al. [25,37], Schieberle et al. [38], Wang et al. [39], Wong et al. [40], and Zabetakis et al. [40].

**Table 1 molecules-24-03104-t001:** Furaneol content of Msalais.

Msalais Sample	°Brix	pH	Alcohol (%)	Furaneol		
				Content (mg/L)	OAV1	OAV2
ms1	9.7 ± 0.0c	3.96 ± 0.01i	12.3 ± 0.1gh	81.84 ± 0.00d	818	16,368
ms2	12.7 ± 0.0f	3.66 ± 0.03e	9.0 ± 0.1bc	56.51 ± 0.07b	565	11,301
ms3	11.8 ± 0.1e	3.61 ± 0.01d	8.8 ± 0.6ab	68.20 ± 0.41fc	682	13,639
ms4	13.0 ± 0.1g	3.10 ± 0.01a	9.5 ± 0.5cd	63.50 ± 0.28c	635	12,700
ms5	10.8 ± 0.0d	3.15 ± 0.03b	8.8 ± 0.0ab	27.59 ± 0.49a	276	5518
ms6	11.0 ± 0.0de	4.00 ± 0.01j	10.6 ± 0.2e	102.73 ± 0.63f	1027	20,545
ms7	10.4 ± 0.1cd	3.56 ± 0.01c	12.1 ± 0.0g	53.42 ± 0.21b	534	10,684
ms8	7.8 ± 0.0ab	3.85 ± 0.01g	9.7 ± 0.1d	78.14 ± 0.54d	781	15,629
ms9	14.3 ± 0.0h	3.73 ± 0.01f	8.1 ± 0.1a	83.17 ± 0.71d	832	16,634
ms10	7.0 ± 0.1a	3.93 ± 0.01h	9.4 ± 0.1c	117.6 ± 0.24g	1176	23,520
ms11	9.0 ± 0.0bc	4.09 ± 0.01k	11.2 ± 0.2f	91.02 ± 0.85e	910	18,204
ms12	10.3 ± 0.0cd	3.93 ± 0.01h	10.4 ± 0.1de	83.66 ± 0.36d	837	16,732
ms13	8.3 ± 0.0b	3.93 ± 0.01g	10.6 ± 0.1e	98.47 ± 0.70f	985	19,693

The differences between values were significant except for the ones indicated by different lowercase letters (one-way ANOVA, *p* < 0.05, Tukey’s test, triplicate for per wine); OAV, odor activity value. OAV1 was calculated by dividing the concentration of furaneol by its odor threshold in water (0.1 mg/L) [20]. OAV2 was calculated by dividing the concentration of furaneol by its odor threshold in 10% hydroalcoholic solution at pH 3.2 (5 μg/L) [35].

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
