# Peer review of "Levels of Furaneol in Msalais Wines: A Comprehensive Overview of Multiple Stages and Pathways of Its Formation during Msalais Winemaking"

_molecules, 2019, doi:10.3390/molecules24173104_

Round 1

Reviewer 1 Report

The study conducted by the authors’ deals with the furaneol content of Msalais specifically its content, stages and pathways of formation during Msalais-making which is nicely presented under Discussion part. The presented findings may help with the Msalais technology improvement by using furaneol as an important indicator of wine quality. The study conducted by the authors is appropriate and according to the scope of the journal, and may be accepted after minor corrections.

Line 92           define OAV

Line 96           “…flavor…”

Line 106         Pls define here and/or under figures i.e. tables all the abbreviation used such are sgj, ogj, HMMF, FaQR/FaEO etc. to enable easier reading

Page 7, Figure 3         How is furaneol bonded? Alpha or beta?

Line 385         What is the LOD and LOQ of the method?

Author Response

The study conducted by the authors’ deals with the furaneol content of Msalais specifically its content, stages and pathways of formation during Msalais-making which is nicely presented under Discussion part. The presented findings may help with the Msalais technology improvement by using furaneol as an important indicator of wine quality. The study conducted by the authors is appropriate and according to the scope of the journal, and may be accepted after minor corrections.

Line 92           define OAV

Thank you for the comment. “OAV” is now clearly defined as theodor activity value”.

Line 96           “…flavor…”

Thank you for the comment. We have made the suggested correction.

Line 106         Pls define here and/or under figures i.e. tables all the abbreviation used such are sgj, ogj, HMMF, FaQR/FaEO etc. to enable easier reading

Thank you for the comment. All of the abbreviationsare now defined in figure legends. HMMF and  FaQR/FaEO are defined in figure

Page 7, Figure 3         How is furaneol bonded? Alpha or beta?

It has been reported that furaneol is bonded to β-d-glucoside and its 6'-O-malonyl derivative (Krammer, Takeoka and Buttery, 1994; Guedes de Pinho and Bertrand, 1995; Schwab, 2013). This information is now provided in the manuscript text in line 252-253

Guedes de Pinho, P. and Bertrand, A. (1995) ‘Analytical determination of furaneol (2,5-dimethyl1-4-hydroxy-3(2H)-furanone). Application to differentiation of white wines from hybrid and various Vitis vinifera cultivars’, American Journal of Enology and Viticulture, 46(2), pp. 181–186. Available at: http://www.ajevonline.org/content/46/2/181.short.

Krammer, G. E., Takeoka, G. R. and Buttery, R. G. (1994) ‘Isolation and Identification of 2,5-Dimethyl-4-hydroxy-3(2H)-furanone glucoside from tomatoes’, Journal of Agricultural and Food Chemistry, 42(8), pp. 1595–1597. doi: 10.1021/jf00044a001.

Schwab, W. (2013) ‘Natural 4-hydroxy-2,5-dimethyl-3(2h)-furanone (furaneol®)’, Molecules, 18(6), pp. 6936–6951. doi: 10.3390/molecules18066936.

Line 385         What is the LOD and LOQ of the method?  

In our previous work (another paper accepted by 食品研究与开发, Food Research And Development ), the LOD and LOQ of the method are 0.076 mg/l. This information is now provided in the manuscript text

Reviewer 2 Report

The authors of this paper report on furaneol concentrations in Mslais wine, a Chinese wine produced from boiled grape juice. The authors report that furaneol is extraordinarily high (27-117 mg/L), one to two orders of magnitude above any previous values reported in wine. These numbers are even more extraordinary considered that conventional table wines produced from ‘Heitianhong’ (the grape used Mslais) is described as ‘neutral’. The authors then quantify furaneol in various systems to conclude that both enzymatic (microbial) pathways and non-enzymatic pathways contribute to the high furaneol of Mslais. Although Mslais is widely available commercially, the general concept of boiling juice to make a concentrated must prior to fermentation is ancient and widespread, and the results could be of broader interest.

Unfortunately, this paper is unacceptable for publication because the authors’ method for measuring furaneol has not been validated. The authors do not specify which wavelength they used for quantification, but presumably it was close to the UV/VIS absorbance max of 292 nm. For example, Walsh, et al J. Agric. Food Chem. 1997, 45, 1320−1324 reported a sizable interference for furaneol in orange juice subjected to accelerated aging, and describe an approach for correcting for the interference. The potential for interferences in wine, with its higher polyphenol content, seems likely to be even higher.

In support of my concerns, the authors report that even the original ‘nf’ treatment, or ‘Naturally-fermented non-heated grape juice’ contained ~1 mg/L furaneol. This is a concentration observed only in certain highly aromatic non-vinifera wines like Muscadine, not in ‘neutral’ wines (authors’ description) like Heitianhong.

To make the extraordinary claim that Mslais wine has 27-117 mg/L furaneol the authors need to provide convincing evidence that they are not measuring interferences by HPLC. This could involve showing UV/VIS spectra for eluting peaks and an authentic standard, or running a subset of samples by an alternative method (such as LC-MS, or LC-UV/VIS on a different column).

Author Response

The authors of this paper report on furaneol concentrations in Mslais wine, a Chinese wine produced from boiled grape juice. The authors report that furaneol is extraordinarily high (27-117 mg/L), one to two orders of magnitude above any previous values reported in wine. These numbers are even more extraordinary considered that conventional table wines produced from ‘Heitianhong’ (the grape used Mslais) is described as ‘neutral’. The authors then quantify furaneol in various systems to conclude that both enzymatic (microbial) pathways and non-enzymatic pathways contribute to the high furaneol of Mslais. Although Mslais is widely available commercially, the general concept of boiling juice to make a concentrated must prior to fermentation is ancient and widespread, and the results could be of broader interest.

Thank you for the comment.

Unfortunately, this paper is unacceptable for publication because the authors’ method for measuring furaneol has not been validated. The authors do not specify which wavelength they used for quantification, but presumably it was close to the UV/VIS absorbance max of 292 nm. For example, Walsh, et al J. Agric. Food Chem. 1997, 45, 1320−1324 reported a sizable interference for furaneol in orange juice subjected to accelerated aging, and describe an approach for correcting for the interference. The potential for interferences in wine, with its higher polyphenol content, seems likely to be even higher.

Thank you for the comment. Furaneol was detected at 286 nm and this information is now provided in section 4.5.

In support of my concerns, the authors report that even the original ‘nf’ treatment, or ‘Naturally-fermented non-heated grape juice’ contained ~1 mg/L furaneol. This is a concentration observed only in certain highly aromatic non-vinifera wines like Muscadine, not in ‘neutral’ wines (authors’ description) like Heitianhong.

Hetianhong, as a local grape, is frequently consumed as a table grape and it is often used to ferment Msalais. Indeed, fermentation of white wine from Hietianhong grape in the laboratory has been reported (Yang et al., 2011). In that paper, the aromatic compounds from grape such as terpenes etc. were unidentified in the wine. If the Hetianhong grape would be an aromatic grape, some aromatic compounds from the grape would be identified in the resultant wine. We therefore respectfully suggest that the descriptor “neutral” is appropriate for the Hetianhong grape.

杨继红, 来疆文, & 高畅. (2011). 新疆和田红干白葡萄酒香气成分分析. 中国酿造, (11), 163-166.( Ji-hong Yang, Wen-jiang Lai, Chang Gao. Analysis of aroma components in dry white wine of Xingjiang Hetianhong[J];China Brewing;2011,11, 163-166)

To make the extraordinary claim that Mslais wine has 27-117 mg/L furaneol the authors need to provide convincing evidence that they are not measuring interferences by HPLC. This could involve showing UV/VIS spectra for eluting peaks and an authentic standard, or running a subset of samples by an alternative method (such as LC-MS, or LC-UV/VIS on a different column).

Indeed, validation of the method for furaneol quantification has been presented in another paper (accepted by 食品研究与开发 Food Research And Development).

We agree that the results of the current study could be confirmed by using an alternative method. However, that seems unnecessary in the case of furaneol, a compound that is abundant in Msalais because HPLC was shown to be adequate means for the identification and quantification of low levels of furaneol in some foods  (Krammer, Takeoka and Buttery, 1994; SANZ, PÉREZ and RICHARDSON, 1994; Walsh, Rouseff and Naim, 1997; Haleva-toledo et al., 1999; Wang and Ho, 2008)

Haleva-toledo, E. et al. (1999) ‘Effects of L-Cysteine and N-Acetyl-L-cysteine on 4-Hydroxy-2,5-dimethyl-3(2H)-furanone (Furaneol), 5-(Hydroxymethyl)furfural, and 5-Methylfurfural Formation and Browning in Buffer Solutions Containing either Rhamnose or Glucose and Arginine’, 3, pp. 4140–4145.

Krammer, G. E., Takeoka, G. R. and Buttery, R. G. (1994) ‘Isolation and Identification of 2,5-Dimethyl-4-hydroxy-3(2H)-furanone glucoside from tomatoes’, Journal of Agricultural and Food Chemistry, 42(8), pp. 1595–1597. doi: 10.1021/jf00044a001.

SANZ, C., PÉREZ, A. G. and RICHARDSON, D. G. (1994) Simultaneous HPLC Determination of 2,5Dimethyl4hydroxy3 (2H)Furanone and Related Flavor Compounds in Strawberries, Journal of Food Science, 59(1), pp. 139141. doi: 10.1111/j.1365-2621.1994.tb06918.x.

Schwab, W. (2013) ‘Natural 4-hydroxy-2,5-dimethyl-3(2h)-furanone (furaneol®)’, Molecules, 18(6), pp. 6936–6951. doi: 10.3390/molecules18066936.

Walsh, M., Rouseff, R. and Naim, M. (1997) ‘Determination of Furaneol and p-Vinylguaiacol in Orange Juice Employing Differential UV Wavelength and Fluorescence Detection with a Unified Solid Phase Extraction’, J. Agric. Food Chem., 45, p. 1320. doi: 10.1021/jf960435z.

Wang, Y. and Ho, C. T. (2008) ‘Formation of 2,5-dimethyl-4-hydroxy-3(2H)-furanone through methylglyoxal: A maillard reaction intermediate’, Journal of Agricultural and Food Chemistry, 56(16), pp. 7405–7409. doi: 10.1021/jf8012025.

Reviewer 3 Report

Please see attached file for comments/suggestions

Reviewer 4 Report

This is an interesting manuscript that determines the levels of Furaneol in a traditional wine of China. However there is some issues that have to be addressed. The authors said that the levels of Furaneol can be used to determine the quality of wine. But in the introduction and conclusion section it is not clear the relationship between Furaneol and quality. I mean, high levels are indicators of bad or good quality? It is important to better understanding this point to focalize the need to quantify this compound in wine. For me it is not clear in the manuscript.

Title: Please modify the title of the manuscript. I would delete the Surprisingly word from the title. I suggest to add traditional fermented wine Msalais and I would be also interesting to delete the coma from the tittle.

Line 98: Please correct the table 1. Some numbers are in italics and others in roman. This is not the best way to indicate significant differences. Please use other symbols. Also, the mean and standard deviation have to have the same significant numbers. In pH values and furaneol content, I recommend only two significant numbers. I also recommend to carry out some statistical analysis and add the results in table 1. For example to evaluate significant differences in values of pH, alcohol and furaneol content. Is there a correlation between these values?

Line 116: Finial???

Line 115-134: It is a bit difficult to read this section. Please modify. It is only a repetition of numbers. Also, the name of the experiments are very difficult to understand. Please reconsider this and change the name of the experiments for a better understanding.

Line 152: A lot more….Please rewrite

Line 159: Please use the same significant numbers, not two numbers in some occasions and three numbers in other occasions. In a manuscript is very important to be consistent with the format. These kinds of mistakes indicates a bad review for authors before submit the manuscript.

Figure 2: Please, in the figure write the meaning of EH and AH. I suppose that it is enzymatic hydrolosys and acidic hydrolysis.

Line 189-190: But high furaneol is better than low furaneol levels or to the contrary.

Discussion section is a bit repetitive and in some occasions have the same information than in introduction section.

Line 300-317: This paragraph is like a conclusion section. Please modify this paragraph and add some of that information in conclusion section.

Line329: It should be important to kwon the time since the elaboration of wines.

Line 342-346: Maybe the authors could add an scheme for better visualization of the experiment.

Table 2: The same as before mentioned. The name of the experiments are a bit difficult to understand.

Line 380: To what absorbance was the furaneol detected? An internal stardand was used?

Author Response

This is an interesting manuscript that determines the levels of Furaneol in a traditional wine of China. However there is some issues that have to be addressed. The authors said that the levels of Furaneol can be used to determine the quality of wine. But in the introduction and conclusion section it is not clear the relationship between Furaneol and quality. I mean, high levels are indicators of bad or good quality? It is important to better understanding this point to focalize the need to quantify this compound in wine. For me it is not clear in the manuscript.

Furaneol is one of the key aromatic compounds that contribute to the caramel odor of alcoholic beverages(Ferreira et al., 2002), also Msalais (submitted). Hence, the higher the level of furaneol, the stronger the caramel odor of wine. An extremely strong caramel odor diminishes the aromatic and flavor complexity of wine( Zhu et al., 2013). On the other hand, a moderate furaneol content enhances the dried fruity odor of wine [submitted]. Therefore, extremely high levels of furaneol impair the quality of wine, while moderate furaneol levels support the quality of wine. However, the specific range of furaneol content also depends on other complex flavor compounds and aromatic compounds, and the quality of wine designed by enologists. This issue undoubtedly requires further study.

In the Discussion in the revised manuscript, to clarify this issue, we now say “The higher the level of furaneol, the stronger the caramel odor of wine. However, while moderate furaneol content enhances the quality of wine, extremely high furaneol content and strong caramel odor diminish the aromatic and flavor complexity of wine, impairing the wine quality”. (lines 312-314)

Ferreira, V. et al. (2002) ‘Chemical characterization of the aroma of Grenache rosé wines: Aroma extract dilution analysis, quantitative determination, and sensory reconstitution studies’, Journal of Agricultural and Food Chemistry, 50(14), pp. 4048–4054. doi: 10.1021/jf0115645

.Zhu, L. et al. (2013) ‘Quantitative Descriptive Analysis of Sensory Characteristics of Musalais from A’wati, Xinjiang’, Food Science, 34(1), pp. 38–44.

Title: Please modify the title of the manuscript. I would delete the Surprisingly word from the title. I suggest to add traditional fermented wine Msalais and I would be also interesting to delete the coma from the tittle.

The Title now reads “Levels of Furaneol in Msalais Wines: A Comprehensive Overview of Multiple Stages and Pathways of Its Formation during Msalais Winemaking”

Line 98: Please correct the table 1. Some numbers are in italics and others in roman. This is not the best way to indicate significant differences. Please use other symbols. Also, the mean and standard deviation have to have the same significant numbers. In pH values and furaneol content, I recommend only two significant numbers. I also recommend to carry out some statistical analysis and add the results in table 1. For example to evaluate significant differences in values of pH, alcohol and furaneol content. Is there a correlation between these values?

As suggested, we performed statistical analysis of the data presented in Table 1, and have now denoted significant differences by lowercase letters.

There are no strict correlations between pH, alcohol, and °Brix in Msalais. Theoretically, in wine, the higher the sugar content (°Brix), the lower the alcohol content. However, some enologists occasionally add alcohol to wine to stop the fermentation or sugar to boost the fermentation.

Line 116: Finial???

The word has now been revised to “end”.

Line 115-134: It is a bit difficult to read this section. Please modify. It is only a repetition of numbers. Also, the name of the experiments are very difficult to understand. Please reconsider this and change the name of the experiments for a better understanding.

As suggested, we have removed most of the repeating numbers in the indicated passage. The sample names are now clearly described in the experimental section (section 4.3) and in the legend to Figure 1.

Line 152: A lot more….Please rewrite

The sentence now reads: “More furaneol was produced…”

Line 159: Please use the same significant numbers, not two numbers in some occasions and three numbers in other occasions. In a manuscript is very important to be consistent with the format. These kinds of mistakes indicates a bad review for authors before submit the manuscript.

Thank you for the comment. Accordingly, we are now using the same significant numbers for the same measurements in the manuscript.

Figure 2: Please, in the figure write the meaning of EH and AH. I suppose that it is enzymatic hydrolosys and acidic hydrolysis.

Thank you for the comment. We now explain the meaning of EH and AH in the legend to Figure 2.

Line 189-190: But high furaneol is better than low furaneol levels or to the contrary.

As explained in our response to your first comment, high levels of furaneol can overpower the aroma of wine, negatively affecting the quality of wine. That point is now explained in the Discussion.

Discussion section is a bit repetitive and in some occasions have the same information than in introduction section.

We have removed repetitive sentences from the Discussion, as suggested. 

Line 300-317: This paragraph is like a conclusion section. Please modify this paragraph and add some of that information in conclusion section.

This paragraph showcases the possible applications of the findings of the study. We provide a separate Conclusions section on the main findings of the study as section 5.

Line329: It should be important to kwon the time since the elaboration of wines.

Thank you for the comment. We now clearly explain in section 4.2 that that the wines were made in 2017.

Line 342-346: Maybe the authors could add an scheme for better visualization of the experiment.

We have now described the experimental flow for each model (section 4.3).

Table 2: The same as before mentioned. The name of the experiments are a bit difficult to understand.

The sample names are now explained in section 4.3 and in the legend to Figure 1,2

Line 380: To what absorbance was the furaneol detected? An internal stardand was used?

Furaneol was detected at 285 nm using a UV detector; purified furaneol was used as an external standard, without an internal standard.

Round 2

Reviewer 3 Report

Suggestion for title change:

Furaneol Levels in Msalais Wines: A  Comprehensive Overview of Multiple Stages and Pathways of Its Formation During Winemaking

Reviewer 4 Report

The authors have improved the quality of the manuscript according the comments of the reviewers. 

A final comment:

The names of the experiments are hard to understand.